# Acceleration of Trading System Back End with FPGAs Using High-Level Synthesis Flow

**Sunil Puranik [1,*], Mahesh Barve [1], Swapnil Rodi [1] and Rajendra Patrikar [2]**

[1]  Tata Consultancy Services, Pune 411057, India
[2]  Visvesvaraya National Institute of Technology, Nagpur 440012, India
*  Correspondence: sunilsavitap@students.vnit.ac.in

**Abstract:** FPGA technology is widely used in the finance domain. We describe the design of a financial trading system order processing component using FPGAs, implemented with high-level synthesis (HLS) flow. The order processing component is the major contributor to increased delays and low throughput in the current software implementation of trading systems. The objective of FPGA implementation is to reduce the latency of order processing and increase the throughput of trading systems as compared to software implementation. Our design is one of the first attempts to speed up order processing in a trading system using FPGA technology and HLS flow. HLS was used in implementing the design for higher productivity and faster turnaround time. The design shows orders of magnitude of improvement in performance indicating that more complex FPGA systems could be designed using HLS. We obtained more than 2X of an advantage in order processing speed and a reduction in latency with FPGA technology. Moreover, we gained a 4X advantage in terms of productivity using HLS.

**Keywords:** accelerator architectures; field programmable gate arrays; high-level synthesis; system performance; TCPIP

## 1. Introduction

Securities trading systems involve the processing of orders that are generated by end users. These orders are typically placed at a rate of 1 million orders per second and are expected to be processed at very low latencies. Since all the components of a trading system such as order validation, lookups, and order matching are implemented in software in a traditional trading system, the order processing rate is low and the latencies of order processing are high. Furthermore, physical network delays and TCP/IP stack delays add to the software delays, resulting in high latencies and low order processing throughput. So, the idea is to speed up the operation of trading systems by migrating the functionality of trading system components including order validation, order matching, lookups, and TCP/IP stack processing from software to hardware.

### 1.1. Use of FPGAs for Accelerating Trading Systems

The number of trading systems is not very large (around 60 stock exchanges in the world) [1] and trading systems contain modules that need frequent reconfigurations of their algorithms as well as parameters. For example, business logic in stock exchanges requires frequent changes, such as the addition of multi-leg order commands (a multi-leg options order refers to any trade that involves two or more options that are completed at once). Since volumes are low and functionality requires frequent changes, the use of ASIC technology is not justified for trading systems acceleration, and reconfigurable computing devices such as FPGAs are the best choice for the acceleration of trading systems.

FPGAs [2] are increasingly receiving traction in the field of financial processing where there is a need for frequent changes in business logic and operating parameters such as the

load and number of securities to be traded. Added to this, there could be a need for adding newer algorithms to the existing system to make it more intelligent.

The development time taken by classical VHDL/Verilog-based flows is very long and productivity is low [2]. There has been a search for alternate flows which can reduce the development time. High-level synthesis (HLS) [3–6] provides a level of abstraction higher than Verilog and VHDL. HLS can be used to describe algorithms in C/C++ and convert them to digital circuits [3]. Additionally, the productivity gained by HLS is orders of magnitude greater than by traditional methods [7]. HLS is supported through its products by a number of VLSI vendors such as Vivado HLS by Xilinx [8], HLS by Intel Altera [9], and Catapult by Mentor [10]. All these products provide tools for writing code in high-level languages such as C/C++/System C and converting them to Verilog/VHDL.

HLS has been traditionally used for implementing algorithmic workflows making use of C language. HLS finds use in domains such as image processing and high-frequency trading (HFT). Boutros et al. [11] described the usage of HLS for designing an HFT system. Here, we use HLS for speeding up the trading system itself.

As shown in Figure 1 below, the trading system environment consists of users/traders submitting trade requests and a trading system which is located in the stock exchange. While HFT trading provides high-speed processing for users submitting trade requests and sits on the user side, our objective in this paper is to accelerate the trading system itself. This paper uses HLS to migrate the functionality of trading system components which are currently implemented in software, to FPGA hardware. This is performed to reduce latency and increase throughput.

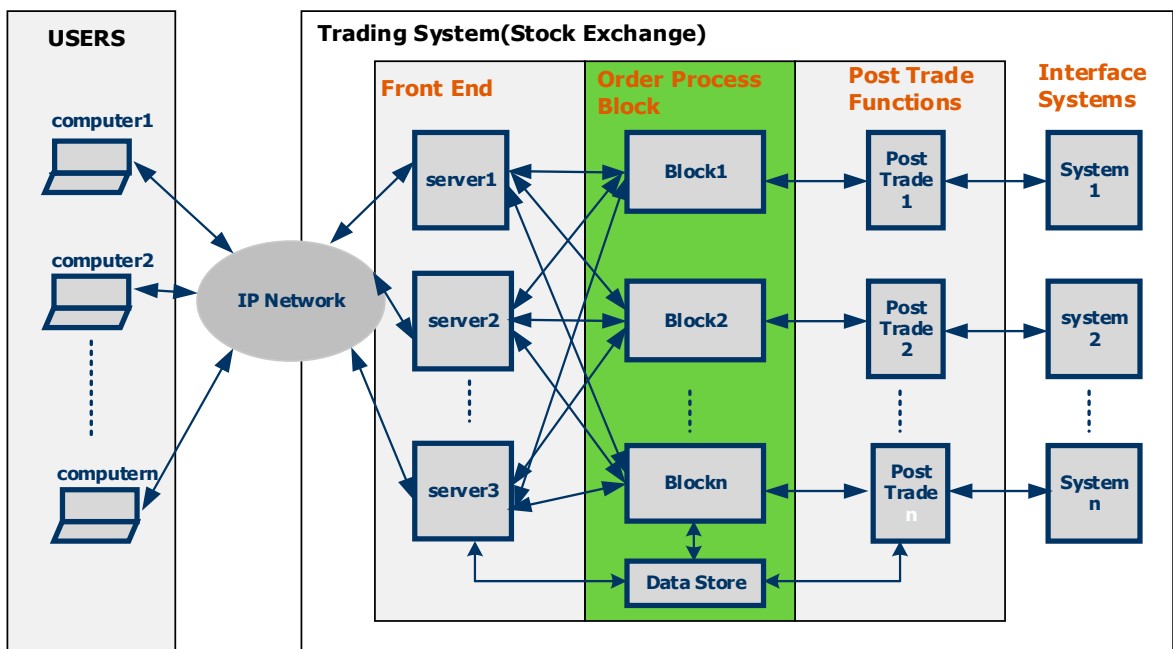

**Figure 1.** Trading system architecture (the block being accelerated is shown in green color).

*1.2. Study Contributions*

The main contributions of this paper are as follows:

- To increase the order processing rate of the trading system from 1 million orders/sec (achieved with software implementation) to 2 million orders/sec with FPGA technology.
- To reduce the latency of order processing commands from 1 microsecond (achieved with software implementation) to less than 500 ns. The throughput and latency numbers for software implementation have been taken from a large stock exchange.

- Additionally, as an important feature of our design, to optimize the use of block RAM (BRAM) which is a fast on-chip memory inside the FPGA, by the innovative design of the data structures.
- Through, this design, to also increase the throughput of the UPDATE command by 30–40% using pipelined execution as explained in later sections.

We describe how HLS was used for implementing the three main commands INSERT, UPDATE and DELETE in the trading system back end.

This paper is organized as follows. In Section 2 we describe the related work conducted in this field. Section 3 describes the general architecture of a trading system. In Section 4, we describe the problem statement. Section 5 describes the data structures used in the design and design implementation. Finally, Section 6 describes the performance numbers, followed by Section 7 on the pipelined execution of the UPDATE command, and Section 8 with the Conclusion and Future Work.

## 2. Related Work

There have been many examples in the literature of FPGAs being used for accelerating financial systems, databases, as well as network protocols. They have found use in high-frequency trading (HFT) [12–14]. These are optimized to achieve the lowest possible latency for interpreting market data feeds. FPGA acceleration for HFT has been described in [15]. FPGA implementation using HLS for HFT has been described in [11]. The study in [16] describes the design of a hardware accelerator to speed up the data filtering, arithmetic, and logical operations of a database. The study in [17] describes the acceleration of a TCP/IP stack with an FPGA. However, after an extensive literature review, we could not find any related previous work that describes the acceleration of a trading system front end and back end with an FPGA.

Trading systems have traditionally existed within the software. Trading system software is very much multi-threaded and is usually found in Linux OS [18,19]. The software makes use of hardware features such as pipelining and multicore technologies. There have been very few instances of the use of FPGAs for a complete trading system back end. A very well-known example of the deployment of FPGAs is in the London Stock Exchange [20]. The system promises extensibility and reconfigurability.

There is very little literature regarding the internal architecture of securities trading systems. This is because these details are mainly proprietary in nature. Moreover, there are very few companies in this field, and revealing the internal architecture could dent their competitive advantage. Hence, the architecture details are not published by the trading system developer firms. Due to this, we were not able to compare the performance of an FPGA-based system to other systems. However, we compared the performance of our system to existing software-based systems.

Our paper describes a trading system accelerator design. FPGAs provide a lot of flexibility that can be exploited by programmers and hardware designers to build accelerators. In data analytics, FPGAs are suited for repetitive tasks. They have been incorporated into platforms such as Microsoft's Project Brainwave [21], the Swarm64 Database Accelerator [22], Postgres [23], the Xilinx Alveo Data Center Accelerator [24], and Ryft [25]. Key–value stores [26] have also been accelerated using FPGAs. Also, FPGAs have become a good option for accelerating databases [27].

## 3. Trading System Architecture at a High Level

The architecture of the system is depicted in Figure 1. It consists of a number of users/traders connected to the trading system using an IP-based network. These traders are outside the premise of the trading system. The trading system itself consists of three components:

1. Front end
   a. Connects the traders to an IP network.
   b. Accepts orders from users.

     c.   Performs validations.

2.    Back end (order processing block)

     a.   Performs the function of order matching, i.e., matching sell orders with buy orders and vice versa. It maintains the database of the sell and buy orders received from users and executes commands to perform order matching.

     b.   Connects to the front end via an Ethernet IP network.

3.    Post trade block Once trading is complete, the post trade block performs functions such as:

     a.   Journaling;

     b.   Recordkeeping;

     c.   Sending a response back to a user on an IP network.

As stated above, our objectives are

1.    To reduce the latency of order processing;
2.    To increase the throughput by implementing order processing functions in FPGA hardware.

Both the front end and order processing block (back end) functions, shown in Figure 1, are implemented in the FPGA using a PCIe-based front end processor board and back end processor board. This architecture is shown in Figure 2.

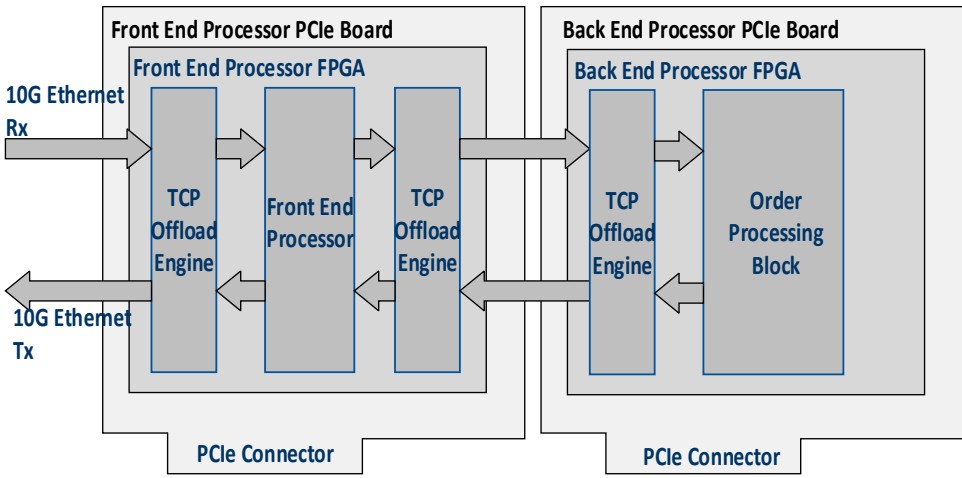

**Figure 2.** Architecture of the trading system implemented in an FPGA.

The users connect to the front end processor board on the 10G Ethernet network and submit trade requests. The front end processor board uses a TCP offload engine (TOE) block to perform TCP/IP processing in hardware to reduce the network latency. It contains the front end processor FPGA. A block diagram of the front end processor FPGA is shown in Figure 3. It contains a TOE which interfaces to users, validations logic, lookups logic, a connections management block, and a TOE for interface to the back end processor board. Validations logic checks the ranges of different fields in the order request submitted by users and verifies that these fields have valid values. Lookups logic performs many lookups to verify that the data in the different fields in the order request matches the master data. The validations and lookups are performed in parallel by FPGA logic to reduce latency. Connections management logic maintains a table of TCP connection IDs against the user IDs and ensures the response from the back end processor board to a user request is sent on the same TCP connection ID on which the order was received. The second TOE performs the function of interfacing with the back end processor FPGA board.

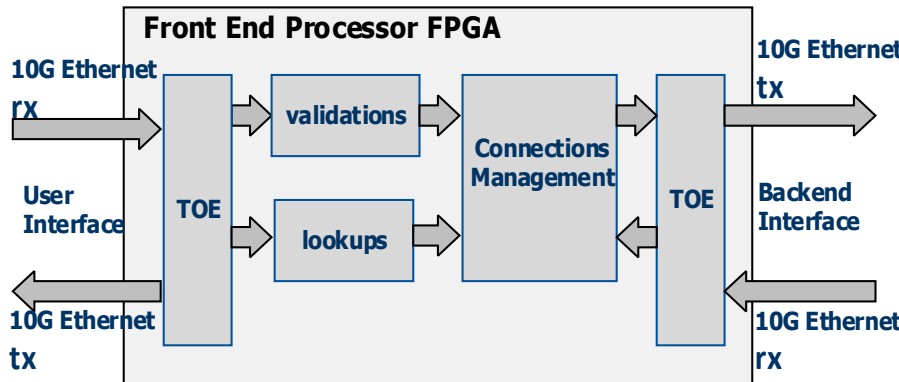

**Figure 3.** Block diagram of the front end processor FPGA.

The back end processor PCIe board connects to the front end processor board on 10G Ethernet and performs order matching functions in hardware to reduce processing latencies. The block diagram of the back end processor board is shown in Figure 4. It consists of a TOE for interface to the front end processor PCIe board and order processing block, which in turn consists of business logic and a command execute block. The business logic matches the sell orders against the buy orders. For example, if there is a buy order for a particular security at a given price and if it matches the sell order with a lesser price for the same security, the trade will be executed. If there is no matching sell order, then the buy order will be inserted into the database. When trade happens, orders are either deleted from the order database or updated in the order database based on order matching quantities. Command execute logic maintains a linked list of orders and performs a deletion, insertion, or update of orders in the order database as described in later sections.

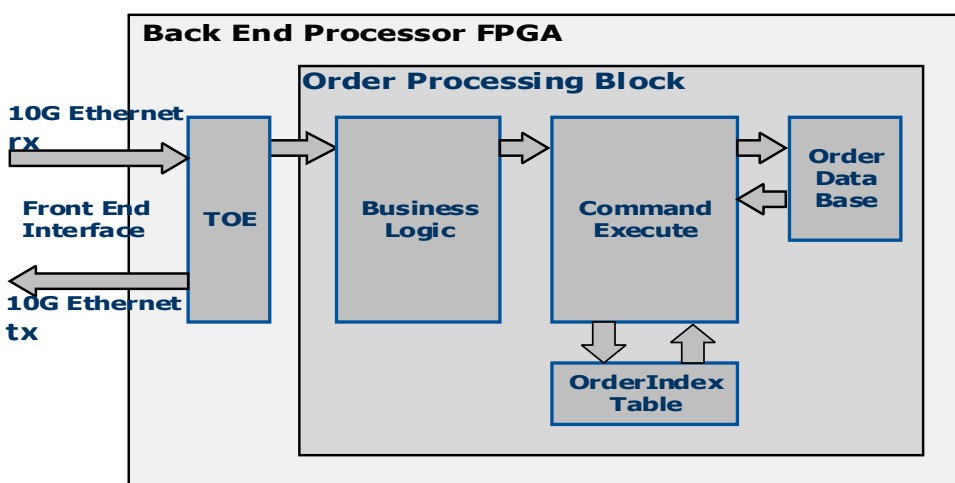

**Figure 4.** Block diagram of the back end processor FPGA.

To reduce network latency, both boards perform TCP/IP processing in hardware using the TOE block. In our implementation, we focus on the order processing block, which is the most critical block in the system in terms of latency experienced for trade requests submitted by users. Order processing involves the implementation of a number of commands out of which INSERT, DELETE, and UPDATE order are the most frequent ones and occur 95% of the time. So, only these three commands are considered for modeling the order processing block:

1. The INSERT order command, when submitted to the trading system back end, requires the incoming order to be placed in the back end order database by appropriately manipulating the internal data structures.

2. The UPDATE order command requires the system to change the price of the already placed order to a new value. Thus, the order data structure stored in the memory is manipulated to indicate the new price to be used. This is the most commonly executed command occurring more frequently than INSERT and DELETE. Hence, it is necessary that the data structures and modules are designed such that the latency of this command is minimized.

3. The DELETE order command requires that the order placed using the INSERT order command is removed from the order database and the order is not manipulated any further.

## 4. Problem Statement

As described in Section 2, the trading system consists of a 10G network, front end, and back end blocks. The total order processing latency consists of three components:

1. Network Latency—This consists of TCP/IP processing delays and wire delays. The TCP/IP processing latency is reduced by implementing TCP/IP processing using a TOE block in hardware as mentioned in Section 3. This reduces latency from 3–4 microseconds (required by the TCP/IP stack implemented in software) to around 100 ns.

2. Front End latency—This consists of delays involved in validations and lookups which are performed by the front end. This is reduced by performing validations and lookups in parallel in FPGA logic.

3. Back End (Order Processing) Latency—This delay is the time required for processing the INSERT, DELETE, and UPDATE commands as described above. This paper describes the implementation of an order processing block in an FPGA using HLS to reduce this latency component.

The INSERT, DELETE, and UPDATE orders form the major chunk of the commands executed in the trading system. Any acceleration of the trading system would require the acceleration of these three commands. Thus, the problem at hand is to increase the throughput of the system and reduce the latency of these transactions. To tackle this problem, newer data structures and algorithms are needed. The constraints for implementing this logic in an FPGA are the on-chip memory (BRAM) and FPGA resources.

## 5. Data Structures

The implementation of the trading system involves the use of the following data structures:

(A) Order Database

The order database stores all the fields and attributes of the order which are placed by the end users. The order structure has all the details needed for processing a transaction. Referring to Figure 5, orders are stored in the order database, which is an array of around one million order structures, stored in Static RAM (a special category of RAM) for fast access. Typical fields in the order structure are the price, time stamp, volume, security identifier, buy/sell flag, OrderID, etc. The offset of the order in the order database is called the OrderIndex and order indexes for all the orders are stored in the order index table shown in Figure 5.

Each order is identified by a unique 32-bit OrderID and this OrderID is used to address the order index table. For example, if orders Order0, Order1, . . . OrderK stored at offsets 0, 1, . . . k, respectively, as shown in Figure 5, have OrderIDs m0, m1, . . . mk, then integer 0 is stored at address m0, integer 1 is stored at address m1 and integer K is stored at address mk in the order index table. This way, using OrderID in an incoming order, the index of the order can be obtained from an order index table lookup and the OrderIndex can be used to locate the order in order database. The order index table is stored in DRAM as OrderID is 32-bit, which requires 4GB of storage.

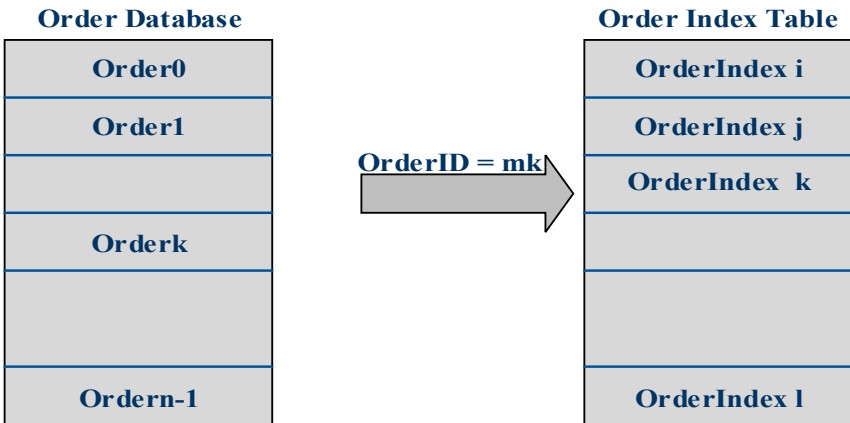

**Figure 5.** Order database and order index table.

(B)　Security Pointers Table

Order nodes are used to store important and frequently accessed information about an order (for example, the price, OrderID, buy/sell flag, etc.). There is one order node corresponding to every order in the order database. For faster access, order nodes are stored in BRAM at the same offset as the corresponding order in the order database. (This offset is the same as the OrderIndex in the order index table). Since order nodes contain only the frequently accessed information about the order, the use of block RAM (BRAM) is optimized. Each security is identified by a unique TokenID. The head pointer to each security linked list is stored in the securities pointer table as shown in Figure 6, at the offset equal to the TokenID. For each security, order nodes for a particular price are stored in a vertical linked list, as shown in Figure 6. They are sorted according to the time stamps. For a given security, there is a vertically linked list of order nodes for each price point. The price point information is stored in a dummy order node and these dummy order nodes are arranged as a horizontally linked list. The dummy order nodes or price points are arranged in the decreasing order of prices for buy orders and in the increasing order of prices for sell orders. Pointers or offsets to the order nodes and dummy order nodes are stored in a free pool, which is accessed as first-in, first-out (FIFO). FIFO stores the offsets of order nodes and dummy nodes. A pointer to the new order node is obtained during an INSERT command execution from the free pool and the pointer is returned to the free pool during the execution of the DELETE command.

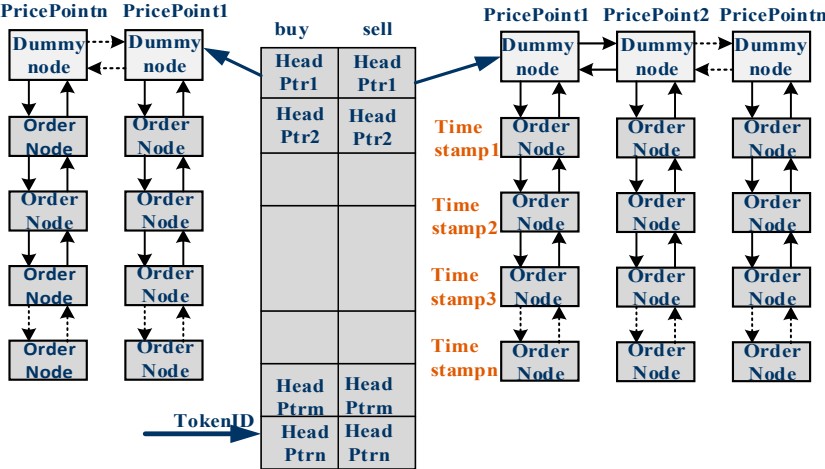

**Figure 6.** Securities pointer table.

## 6. HLS Implementation of the Order Processing Block

A block diagram of the order processing block which executes the INSERT, UPDATE, and DELETE commands is shown in Figure 7 below. The order processing block consists of the following components:

1.  Command Queue—The command queue is used to store the commands delivered on the command interface.
2.  Command Decode Logic—This block reads the commands from the command queue and decodes them. Based on the command code, it calls the different functions in the command execute block to execute the command.
3.  Command Execute Block—This block contains all the subfunctions required to execute the INSERT, DELETE, and UPDATE commands as explained later.
4.  Free Pool FIFO—The Free Pool FIFO stores the pointers to free order nodes and dummy order nodes.
5.  Dummy Order Node Array—The dummy order node array is used to store the linked list of dummy order nodes which contain the price information of buy and sell orders. They are stored in BRAM for faster access.
6.  Order Node Array—The order node array is used to store the linked list of order nodes which contain the frequently accessed information about the orders. These are stored in BRAM for faster access.
7.  Order Index Table—As explained earlier, the order index table is used to store OrderIndex information and is accessed by OrderID. This table is implemented in DRAM.
8.  Command Status Queue—This contains the status of the commands that were delivered on the command interface.
9.  Order Database—The order database contains the orders placed by users of the trading system. It is stored in SRAM for faster access.

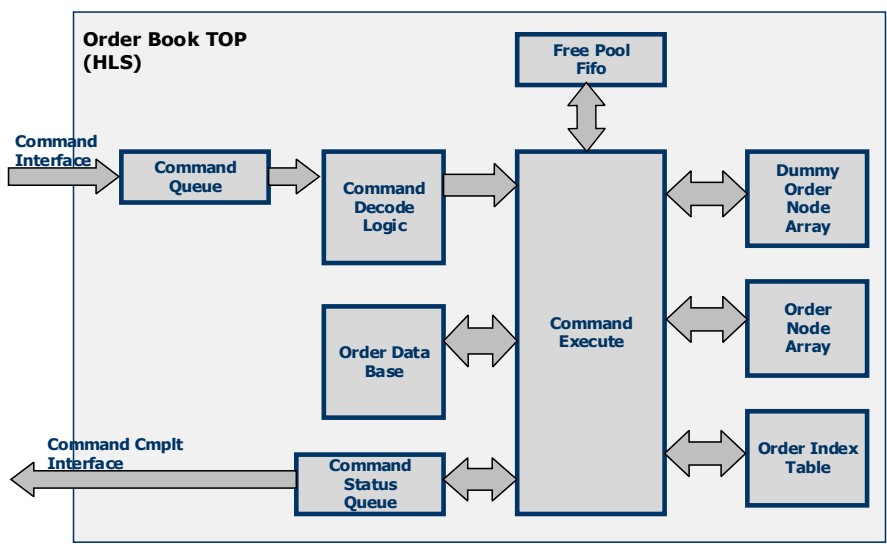

**Figure 7.** Block diagram of the order processing block implemented in HLS.

The INSERT, UPDATE, and DELETE commands delivered over the command interface are stored in the command queue. The command decode logic reads the commands from the queue, decodes the commands, and calls the command execute logic functions to execute the required command. Command execute logic implements HLS functions to read the order node from the free pool FIFO, return the order node to the free pool, insert the order node into the order node array, remove the order node from the order node array, read the OrderIndex from the order index table, and manipulate the pointers for inserting and deleting the order nodes in the order node arrays. The dummy order node array and order node array are implemented as doubly linked lists as shown in Figure 6. As the order

index table is stored in DRAM and the order database is stored in SRAM, which are both off-chip memories, the AXI master interface is used for accessing these data structures. We used the pragma HLS interface m_axi port = ord_ind_arr for implementing the AXI master interface. We also used the pragma HLS interface bram port = ord_nd_arr and pragma HLS interface bram port = dmy_nd_arr for implementing BRAM interfaces for order node and dummy order node arrays, respectively. To implement pipelined operations, pragma HLS pipeline II= n was used. The pipeline pragma was also used to pipeline the loops and obtain a higher frequency operation. Pragma HLS latency = max_value was used for constraining latency values.

Table 1 below shows the HLS pragmas used in the code in tabular form.

**Table 1.** Details of the pragmas used in the HLS code for the order book top.

| Sr. No. | Block Name | Pragma | Value |
|---|---|---|---|
| 1 | Order_book_top | HLS Interface | m_axi port = ord_ind_arr |
| 2 | Order_book_top | HLS Interface | m_axi port = ord_bk_arr |
| 3 | Order_book_top | HLS Interface | Bram port = ord_nd_arr |
| 4 | Order_book_top | HLS Interface | Bram port = dmy_nd_arr |
| 5 | Command Execute | HLS Pipeline | II = 1 |
| 6 | HLS_top | HLS Latency | Max 200 |
| 7 | Command Queue | HLS Stream | Depth = 8 |
| 8 | Command Status Queue | HLS Stream | Depth = 8 |

The steps involved in executing the INSERT, UPDATE, and DELETE commands by the order processing block are described below.

The order processing block (back end logic) decodes the orders received from the front end and takes the following steps during the execution of each of the INSERT, UPDATE, and DELETE order commands:

A. INSERT Order

1. Get the pointer to the new order node (offset of the order node) from the free pool FIFO.
2. Store the order structure in SRAM at the same offset (offset obtained in step 1) as the order node.
3. Make the entry in the order index table in DRAM. Write the offset of the order in DRAM (which is the same as the offset of the order node in BRAM) in the order index table using the OrderID as the address. (The OrderID is received as a part of the order request). Set a flag to indicate that the content of the location is valid.
4. Traverse the dummy order nodes linked list to find the location where the order node corresponding to the new order can be placed based on the price field in the order.

B. UPDATE Order

1. Using the OrderID field in the request as an address, read the OrderIndex (offset of the order in SRAM) from the order index table stored in DRAM memory.
2. Using the OrderIndex, the order node corresponding to the order is accessed.
3. This order node is moved to the new price point position and added at the end of the vertical linked list of order nodes and deleted from the current price position in the vertical linked list under the dummy node corresponding to the old price position. If there is only one order node under the dummy node corresponding to the old price position, the dummy node is deleted and returned back to the free pool FIFO.

4.   If the dummy node corresponding to the new price position does not exist, it is obtained from the free pool and added to the horizontal linked list, and the order node is added at the head of the vertical linked list under the newly added dummy node.

C.   DELETE Order

1.   Using the OrderID field in the request as an address, read the OrderIndex from the order index table stored in DRAM. Reset the flag in the DRAM location indicating that the location content (OrderIndex) is no longer valid.

2.   Locate the order node using the OrderIndex. The order node is at the same offset in block RAM (BRAM) as the order structure in SRAM and this offset is equal to the OrderIndex. Remove the order node from the linked list by manipulating the pointers in the time vertical linked list. If it was the only node under the dummy node, then remove the corresponding dummy node as well. Return the order node back to the free pool FIFO.

The algorithmic steps for executing the INSERT, UPDATE, and DELETE commands are shown in Figure 8 below:

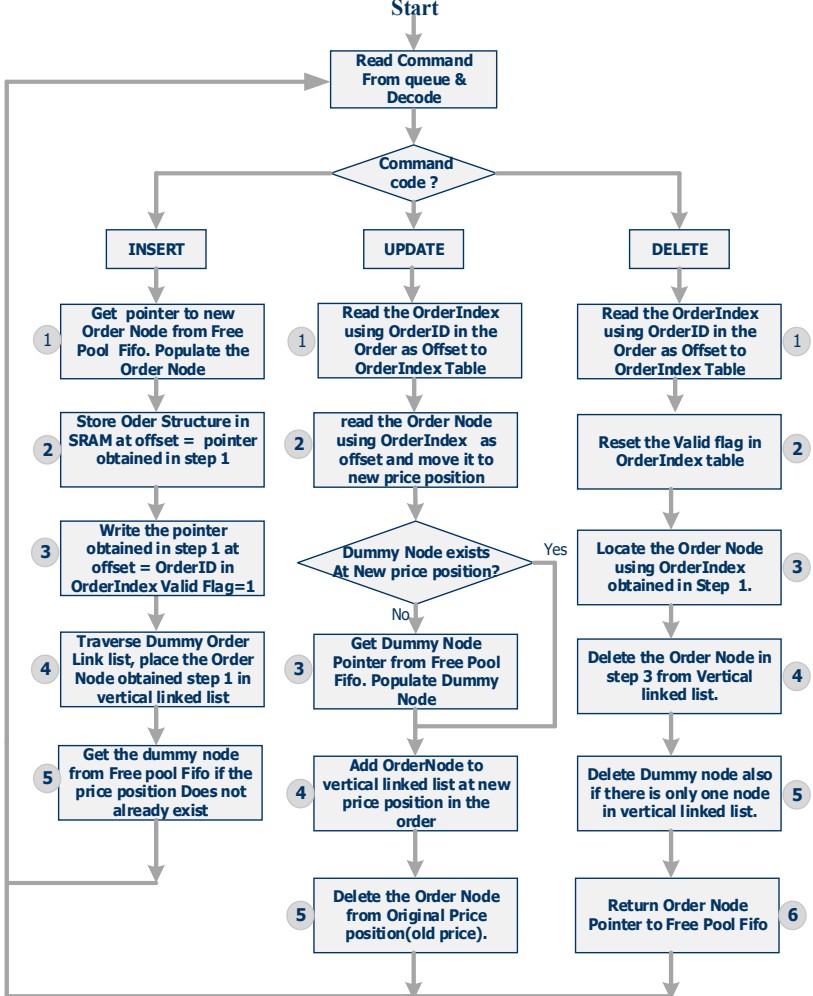

**Figure 8.** Algorithms for the execution of the INSERT, UPDATE, and DELETE commands.

## 7. Performance Numbers

To obtain the performance numbers, we implemented the order processing block in HLS, Verilog, and software. The setup consisted of a Vivado HLS IDE and QuestaSim simulator for SystemVerilog. HLS code was run, first in the Vivado HLS IDE environment to confirm logical correctness. Co-simulation was conducted to understand whether

the generated Verilog also worked correctly. The HLS code was synthesized on a Xilinx Ultrascale+ FPGA board (Virtex UltraScale+ VCU118-ES1 Evaluation Platform with xcvu9p-flga2104-2L FPGA). It was synthesized with a clock cycle of 3 ns. Latency numbers for overall processing (front end + order processing block) were computed for the system using C-RTL co-simulation in Vivado and the use of SystemVerilog simulations under QuestaSim. The software implementation of the front end and the order processing block consisted of C code run on a fault tolerant machine with a Red Hat Linux 7, 16-core CPU (Intel (R) Xenon(R) CPU ES-2667 V3 @ 3.20 GHz), and 256GB of RAM. The data structures used in the software were different since there was no consideration of the block RAM for software implementation. The software uses hashmap and treemap data structures for the order book. As for the synchronization, single-writer principles were followed to avoid locking contentions in the performance critical path. In a few scenarios, compare and swap low latency locks were used. Dedicated isolated cores were assigned to every process to avoid CPU switching. Due to the sequential nature of software, having more CPU cores did not give significant performance improvement.

The performance of the design was measured based on various parameters, namely, resource utilization, latency, and throughput. Overall, the time required for processing one order was around 500 ns for HLS and Verilog, while it was around 1 microsecond for software. Table 2 gives the details of the resource utilization of the final system after synthesis in the xcvu9p-flga2104-2L FPGA. These details of the resource utilization were made available by the Vivado HLS synthesis tool.

**Table 2.** Resource utilization.

| Flip Flops/ Total/ Utilization | LUTs/ Total/ Utilization | Memory kb/ Total/ Utilization |
|---|---|---|
| 10629/ 2364480/ 0.45% | 18769/ 1182240/ 1.58% | 105/ 4320/ 2% |

### 7.1. Setup for Latency and Performance Measurement

A block diagram of the test bench and design under test (DUT) for measuring the latency and performance of different commands is shown in Figure 9. The DUT consists of Verilog code of the order processing block implemented in HLS. The test bench consists of the command generator, command queue, command latency mean execution time calculator, and report generator. The command generator generates a random mix of INSERT, UPDATE, and DELETE commands using SystemVerilog constrained random generation and submits the commands to the DUT on the command interface. The weighted random distributions of different commands are generated using the *dist* operator of SystemVerilog random generation. Commands are also queued into the command queue as they are submitted to the DUT along with the time stamp. Commands are executed by the DUT in the order in which they are submitted. The DUT indicates that command execution is complete with the Cmd_cmplt pulse shown in Figure 9. This signal is used to record the command completion time. Commands are retrieved from the command queue in FIFO order, and the command execution time and command latency are calculated by the mean time and latency calculate block, respectively. This block also maintains the count of how many UPDATE, DELETE, and INSERT commands were executed in a particular test case. The report generator prints the report of the latency and command mean execution time for the INSERT, UPDATE, and DELETE commands.

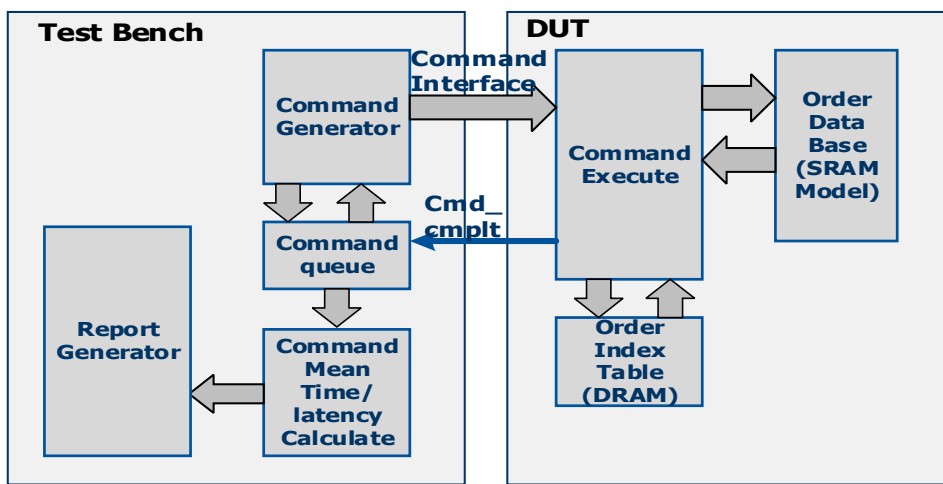

**Figure 9.** Block diagram of the SystemVerilog test bench and DUT.

*7.2. Latency Measurements*

The following were our observations with regard to latency for each of the commands:

1.　DELETE Order The latency for the DELETE order operation remains constant. It does not change based on the order number or the location of the order in the buy/sell linked list.

2.　INSERT Order

The INSERT operation involves traversing the buy/sell dummy order node linked list and placing the incoming order under the appropriate dummy order node. If needed (if the price point does not exist), a new dummy order node may be inserted, and the incoming order placed under this dummy node. Thus, we see that the time for the INSERT order is dependent on the time spent traversing the buy/sell linked list or the number of dummy nodes (hops) that have to be inspected. Thus, latency depends directly on the number of hops.

3.　UPDATE Order

The UPDATE operation involves placing one timestamp node in a new location in the buy/sell linked list. The latency is dependent on the number of hops from the current dummy node location to the new dummy node location.

To compute latency, two types of traffic were generated for the system:

a.　Sequential traffic that gave a fixed sequence of INSERT, UPDATE, and DELETE commands.

b.　Random traffic that gave INSERT, UPDATE, and DELETE commands in some weighted proportion. The proportion was configurable.

These timings have been observed with the QuestaSim SystemVerilog Simulator designed by Mentor Graphics.

*7.3. Atomic Transaction Level Latency*

From the latency test cases, the following latencies (refer to Table 3) have been observed for the INSERT, UPDATE, and DELETE commands. (These numbers were calculated from sequential traffic tests by giving few INSERT, DELETE, and UPDATE commands).

**Table 3.** Latencies of various commands.

| Command Name | Latency (Clocks) | Comment |
|:---:|:---:|:---:|
| INSERT | 52 | This is for inserting one price point after the initial insertion. |
| DELETE | 45 | This timing is constant, irrespective of the number of price points, as expected. |
| UPDATE | 36 | This is for the first UPDATE with one hop. |

### 7.4. Formulae for the Expected Latency

For the INSERT/UPDATE commands, the linked list has to be traversed. The number of dummy nodes between the start and the end node is called hops. After running sequential and random traffic tests, we observed the following relationship between the number of hops and corresponding latency for each command:

- INSERT: Clocks for N hops = 50 + 2xN
- UPDATE: Clocks for N hops = 34 + 2xN
- DELETE: Total number of clocks = 45

N in the above formulae is the number of hops. Here, latency is the number of clock cycles with the clock having a period of 3 ns. These latencies were calculated with C-RTL co-simulation and do not include DRAM access latency and DDR controller latency. As expected, the latency of INSERT and UPDATE was proportional to the number of hops while the DELETE latency was constant irrespective of the number of hops.

### 7.5. Observed Latency under Various Price Depths (Hops)

This study is applicable to UPDATE commands.

Table 4 above shows the latency numbers for 1200 total commands of which the first 300 were inserts and the rest were random where the percentages of INSERT, UPDATE, and DELETE were 10%, 80%, and 10%, respectively.

**Table 4.** Latencies of various depths for UPDATEs.

| Max. Hop | Min. Latency (Clocks) | Max. Latency (Clocks) | Avg. Latency (Clocks) | Std. Dev. Latency (Clocks) |
|:---:|:---:|:---:|:---:|:---:|
| 20 | 70 | 149 | 89 | 10 |
| 30 | 70 | 159 | 96 | 15 |
| 40 | 70 | 165 | 100 | 16 |

Table 5 below has 1200 total commands of which the first 300 were INSERTs and the rest were random where the weights of the INSERT, UPDATE, and DELETE commands entered were 5%, 90%, and 5%, respectively. From the tables, we can conclude that the minimum time was for the first UPDATE. It can be inferred from the table that irrespective of the distribution, the average latency and maximum latency depend on the number of hops, while the minimum latency remains constant as expected.

**Table 5.** Latencies of various hops for UPDATEs.

| Max. Hop | Min. Latency (Clocks) | Max. Latency (Clocks) | Avg. Latency (Clocks) | Std. Dev. Latency (Clocks) |
|:---:|:---:|:---:|:---:|:---:|
| 20 | 70 | 151 | 90 | 10 |
| 30 | 70 | 171 | 97 | 15 |
| 40 | 70 | 169 | 101 | 17 |

*7.6. Throughput*

The throughputs for the INSERT, UPDATE, and DELETE commands were calculated based on the mean time required for the execution of the commands for a given number of hops. The mean time gives the time it takes in ns to execute the command. The throughput of the system was computed under various kinds of loads. The following table, Table 6, depicts the throughput for the INSERT, UPDATE, and DELETE commands. The hops in the table indicate the initial depth of the linked list which vary according to the INSERT and DELETE traffic.

**Table 6.** Throughput of various commands.

| Sr. No. | Command | Hops | Execution Times (ns) | Throughput (commands/sec) |
|---------|---------|------|---------------------|---------------------------|
| 1 | INSERT | 20 | 367 | $2724.79 \times 10^3$ |
| 2 | INSERT | 30 | 361 | $2770.08 \times 10^3$ |
| 3 | INSERT | 40 | 367 | $2724.79 \times 10^3$ |
| 4 | UPDATE | 20 | 270 | $3703.703 \times 10^3$ |
| 5 | UPDATE | 30 | 291 | $3436.4 \times 10^3$ |
| 6 | UPDATE | 40 | 303 | $3300.330 \times 10^3$ |
| 7 | DELETE | 20 | 153 | $6535.9 \times 10^3$ |
| 8 | DELETE | 30 | 162 | $6172.8 \times 10^3$ |

So, the throughput for the command is

$(10^9)$/(mean command execution time) commands per second

Note: This calculation was obtained with traffic from 90% UPDATE, 5% INSERT, and 5% DELETE commands.

The software implementation of the INSERT and UPDATE commands takes 1.5 microseconds (0.67 million commands/sec) while DELETE takes 1.2 microseconds (0.84 million commands/sec). It can be seen that with an FPGA, the throughput of all the commands increased to more than 2 million commands/sec (which is the same as orders/sec). Furthermore, the average latency of the command execution was reduced to around 300 ns.

*7.7. Productivity*

Verilog implementation took 6 months while HLS implementation took 1.2 months with two engineers with experience of 8–10 years.

## 8. Pipelined Execution of UPDATE Command

The execution of the UPDATE command involves two phases: 1. Fetch—In this phase, the OrderIndex of the order which is stored in the order index table is fetched using the OrderID as the address. Since the order index table is stored in DRAM, the fetching of the OrderIndex takes longer compared to BRAM. 2. Execute Phase—This phase involves moving the order node to a new price position, adding at the end of the vertical linked list of order nodes and deleting the order node from the current price position in a vertical linked list under the dummy node corresponding to the old price position. If the fetch and execute phases are carried out sequentially without any overlap, the execution time of the UPDATE command increases, resulting in less throughput for UPDATE. To address this issue, we modified the UPDATE command execution logic such that there is an overlap between the execution of the fetch and execute phases. This was achieved by executing the fetch and execute phases in a pipelined fashion. The fetch logic looks up the order index table using the OrderID as the address and writes the OrderIndex read from DRAM into a first-in, first-out (FIFO) queue. The execute command reads the OrderIndex from the queue and uses it to locate the order in the order database. After locating the order node,

the execute logic moves it to the new position in the horizontally linked list of price nodes. Since the fetching of the OrderID for the next UPDATE command overlaps with the execute phase of the previous UPDATE command, the effective execution time of the UPDATE command is reduced significantly, if the UPDATE commands are received sequentially. Since the percentage of UPDATE command is very high (around 90% of the total commands are UPDATEs), this modification results in a 30–40% increase in the throughput of the UPDATE command as shown in Table 7 below.

**Table 7.** Throughput for various commands with the pipelined execution of UPDATE.

| Command | Hops | Execution Times (ns) | Throughput (commands/sec) |
| :---: | :---: | :---: | :---: |
| INSERT | 20 | 367 | $2724.79 \times 10^3$ |
| INSERT | 30 | 361 | $2770.08 \times 10^3$ |
| INSERT | 40 | 367 | $2724.79 \times 10^3$ |
| UPDATE | 20 | 189 | $5290.703 \times 10^3$ |
| UPDATE | 30 | 218 | $4587.4 \times 10^3$ |
| UPDATE | 40 | 224 | $4464.330 \times 10^3$ |
| DELETE | 20 | 153 | $6535.9 \times 10^3$ |
| DELETE | 30 | 162 | $6172.8 \times 10^3$ |

## 9. Conclusions and Future Work

In this study, we have implemented the order processing block of a trading system with FPGA technology. By migrating the functionality of order processing from software to hardware, we were able to obtain more than 2X of an advantage in throughput and order processing latency was reduced to less than 500 ns. The design was implemented with HLS. HLS methodology is comparatively new and is an emerging technology that is not mature as of yet. However, our observation is that the results of latency and throughput obtained with HLS are very close to Verilog implementation. With HLS, we achieved almost 4X–5X of an improvement in throughput for the INSERT and UPDATE commands compared to software implementation. However, to obtain results close to a highly optimized and efficient Verilog implementation, various optimization techniques need to be tried out as recommended below:

- Using HLS stream variables internally to implement FIFOs and carry out concurrent/overlapped executions of subfunctions of the three commands.
- Using an optimal mix of Verilog and C code in which certain latency and time-critical subfunctions are coded in Verilog, and the rest of the logic is coded in C and implemented in HLS.
- Design under test (DUT) consists of the Verilog implementation of the order processing block. As an alternative approach, the same DUT can be ported on an Intel HLS Compiler, and the results compared with those obtained from Xilinx Vivado HLS.

**Author Contributions:** Conceptualization, S.P. and S.R.; methodology, S.P.; software, M.B.; validation, S.P. and M.B., formal analysis, R.P.; writing—original draft preparation, S.P.; writing—review and editing, S.P. and M.B.; supervision, R.P. All authors have read and agreed to the published version of the manuscript.

**Funding:** Research received no external funding.

**Institutional Review Board Statement:** Not Applicable.

**Informed Consent Statement:** Not Applicable.

**Data Availability Statement:** Study does not report any data.

**Conflicts of Interest:** Authors declare no conflict of interest.

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
