# Peer review of "Acceleration of Trading System Back End with FPGAs Using High-Level Synthesis Flow"

_electronics, doi:10.3390/electronics12030520_

Round 1
Reviewer 1 Report
In this paper, the authors present the acceleration of a high-frequency algorithm. The paper is badly written and does not convey professionalism. In its current state is unsuitable for publication because it carries many problems that should be eliminated if this is going to be published.
1) First of all, the Turnitin plagiarism check reveals that the authors have posted a previous version of it online. There is an 85% similarity with that version. The Editor should check the publication rights and decide if it is compatible to be published also here. The preprint is at: https://www.techrxiv.org/articles/preprint/Acceleration_of_Trading_System_Backend_with_FPGAs_using_High_Level_Synthesis_Flow/19730968
2) The paper lacks many important sections that are found in research papers, or the authors do not devote any text to highlight. Even the abstract is very problematic.
(a) There is no motivation. Why is this important for us?
(b) There is no state-of-the-art. What others are doing?
3) The authors state some facts, but no references are given. For example
(a) "FPGA technology is increasingly finding use in the finance domain", reference
(b) "the order processing rate is low and the latencies of order processing are high.", reference?
(c) "The number of trading systems is not very large", what is large? 10? 1000? 1K? reference?
(d) "need frequent reconfiguration", why? reference?
...and many others. The authors should utilize references. This is a scientific paper, not a personal opinion paper.
4) The authors state some wrong things. For example "FPGAs an ideal candidate for implementation". The ideal candidate is a software implementation on a workstation. The authors stated 2 engineers worked for 1.5 months to develop an HF system.
5) Even though the authors do not utilize required references, on the other hand, they utilize useless references. For example, giving a reference on what is the Linux Operating system [15][16] is totally useless. The authors should utilize references only where needed. Not just put references just for showing some references.
6) Also, the authors do not utilize the proper reference scheme. For example reference [19] in the bibliography is wrong. This is not the correct way of referencing. Almost every reference is not correct according to MDPI quality standards. As a tip: When the authors submit to a new journal, they should read carefully the template and not write without a template.
7) The images have low quality and some of them are unreadable. For example, Figure 1 seems like a copy-paste image from another article and for this reason, the correct scaling is lost. The authors should use their good-quality primary material and not copy-paste images from other sources.
8) The bibliography is very old. There are no articles from the last 3 years, and most of them are over a decade. The authors stated that there are few publications on this subject, but if they google "high-frequency trading using FPGA" a lot of results, some of them from 2022 are found. The authors should carefully study them and compare them.
9) The authors should give the algorithm pseudo description and details. In this current state, the authors say "we accelerated an HF algorithm". They give no name, they give no details (just some very high-level details of the 3 stages). Publishing required the authors to reveal full details for someone else to be able either to repeat this research and verify it or to compare. In this cryptic and non-disclosed form, nobody can benefit from this. In its current state, the only takeaway is "Somebody accelerated an HF algorithm using FPGA". This is a useless take-away conclusion for the research community.
10) Even some figures give the feeling that the authors presented something hastily without trying to increase the quality. For example, Figure 2 has many lines interconnected with each other. If the authors took time to improve it, they would fill the arrows with white color, and the interconnection lines (which makes it a mess) would vanish. When sending a paper for review, the authors should send their best effort that has quality, not a draft version that was written hastily. Another example is Figure 5, which has an arrow not balanced (it is very close to the right table).
11) There are no take-away messages of how exactly the HLS was performed (just stating, we used Vivado HLS is not complete). The authors should specify exactly how they used Vivado HLS, how they inserted the pragmas, how they decided on the hierarchy and the operators, and so on. In this state, no reader will benefit from it and this should not be published, because it violates the good publishing ethics of the research community (together with the very bad quality of the figures and text).
12) Even the comparisons are not well described. Just stating we used 16 cores is useless. What CPU? Why there was latency on 16 cores? Are they very old? If the authors utilized 32 cores it would be up to the speed of FPGA (so this would be better?). The comparisons should be correct and not biased. In their current form, they are biased and they do not give the full picture. Just some numbers that show that FPGA is better. But behind the numbers, I can see that it's not so simple the comparison.
13) Even the algorithm used is not described correctly, and thus I cannot verify the correctness.
14) Even the test measurements are not described correctly. How the testbench was created? How the validation phase passed the tests?
In general, the authors should prepare better this manuscript. This has many problems that prohibit the publication.
Author Response
We would like to thank the reviewer for comments and feedback. Please find our response below inline:
In this paper, the authors present the acceleration of a high-frequency algorithm. The paper is badly written and does not convey professionalism. In its current state is unsuitable for publication because it carries many problems that should be eliminated if this is going to be published.
- First of all, the Turnitin plagiarism check reveals that the authors have posted a previous version of it online. There is an 85% similarity with that version. The Editor should check the publication rights and decide if it is compatible to be published also here. The preprint is at: https://www.techrxiv.org/articles/preprint/Acceleration_of_Trading_System_Backend_with_FPGAs_using_High_Level_Synthesis_Flow/19730968
Author Response – We had submitted this article to IEEE Systems Journal and there was an option there to submit it on the above-mentioned preprint site to get feedback from readers. In fact they mentioned that authors are encouraged to submit to the preprint site. That was the reason it was submitted there. The article however was rejected by IEEE Systems Journal and is not published in any of the journals to this date.
2) The paper lacks many important sections that are found in research papers, or the authors do not devote any text to highlight. Even the abstract is very problematic.
(a) There is no motivation. Why is this important for us?
Author Response – The motivation of this research is to increase the order processing speed of the trading system from 1 million orders/sec to 2 million orders/sec or more. This is mentioned under heading B. Contributions of this paper in the Introduction section. Another motivation is to reduce the latency of order processing from 1 microsecond to less than 500 ns. Current software implementation gives low order processing speed and high latency. Our aim is to increase the throughput and reduce the latency by migrating order-matching functions from software to hardware. The specifications of the latency and order processing speed have been obtained from a major stock exchange in India which is our client.
(b) There is no state-of-the-art. What others are doing?
Author response – We have done an extensive literature survey and have not found any designs implementing trading systems backend with FPGA. As mentioned in the Related Work section, most of the trading systems architectures are proprietary, and revealing this information will affect their competitive advantage. So we have compared the performance with trading system of our client which is implemented in software.
3) The authors state some facts, but no references are given. For example
(a) "FPGA technology is increasingly finding use in the finance domain", reference
Author Response – We have given a reference regarding the financial applications of FPGAs and also described how FPGAs are used in High-Frequency Trading (HFT) in the industry in the Related Work section.
(b) "the order processing rate is low and the latencies of order processing are high.", reference?
Author Response – We have taken the numbers for the order processing rate and latency from a major stock exchange in India which is our client. Since this information is not public, we are not able to furnish any reference. However, the majority of stock exchanges are implemented with software, so delays are high and order processing rate is low due to TCP/IP stack delays, sequential processing, the requirement of look-up operations which cannot be done in parallel with software etc.
(c) "The number of trading systems is not very large", what is large? 10? 1000? 1K? reference?
Author response – This is addressed in the section on Introduction, heading A.
(d) "need frequent reconfiguration", why? reference?
Author response - This is addressed in the section on Introduction, heading A.
...and many others. The authors should utilize references. This is a scientific paper, not a personal opinion paper.
4) The authors state some wrong things. For example "FPGAs an ideal candidate for implementation". The ideal candidate is a software implementation on a workstation. The authors stated 2 engineers worked for 1.5 months to develop an HF system.
Author response- We have made appropriate changes. We have compared effort for Verilog implementation (6 months) with that for HLS (1.2 months) under the productivity heading in section 7 on Performance numbers. As expected, HLS requires much less time compared to verilog
5) Even though the authors do not utilize required references, on the other hand, they utilize useless references. For example, giving a reference on what is the Linux Operating system [15][16] is totally useless. The authors should utilize references only where needed. Not just put references just for showing some references.
Author response- We have given references on Linux because most of the Trading systems implemented with software use Linux Operating System. These can be removed if necessary. Our team has studied the Linux functions used in Trading System operations to check if these can be implemented in hardware.
6) Also, the authors do not utilize the proper reference scheme. For example reference [19] in the bibliography is wrong. This is not the correct way of referencing. Almost every reference is not correct according to MDPI quality standards. As a tip: When the authors submit to a new journal, they should read carefully the template and not write without a template.
Author Response – We have taken care of this by modifying the References section.
7) The images have low quality and some of them are unreadable. For example, Figure 1 seems like a copy-paste image from another article and for this reason, the correct scaling is lost. The authors should use their good-quality primary material and not copy-paste images from other sources.
Author Response – We have redrawn and modified all the figures to take care of this feedback.
8) The bibliography is very old. There are no articles from the last 3 years, and most of them are over a decade. The authors stated that there are few publications on this subject, but if they google "high-frequency trading using FPGA" a lot of results, some of them from 2022 are found. The authors should carefully study them and compare them.
Author Response – We are not trying to accelerate the High-Frequency Trading (HFT) systems. They have been just quoted as examples of applications of FPGA in the finance sector. In fact, HFT systems are used by users who submit trade requests and they are located outside the premise of the trading system. In this research work, we are trying to accelerate the trading system itself with the help of FPGA technology. However, we can change the references later if required and give more recent references.
9) The authors should give the algorithm pseudo description and details. In this current state, the authors say "we accelerated an HF algorithm". They give no name, they give no details (just some very high-level details of the 3 stages). Publishing required the authors to reveal full details for someone else to be able either to repeat this research and verify it or to compare. In this cryptic and non-disclosed form, nobody can benefit from this. In its current state, the only takeaway is "Somebody accelerated an HF algorithm using FPGA". This is a useless take-away conclusion for the research community.
Author Response- We are not clear about the comment. We have not accelerated any “HF algorithm”. If you mean an “HFT algorithm” i.e. High-Frequency Trading algorithm, as stated above, we are not trying to accelerate the HFT system which is located outside the trading system. Our objective is to accelerate the trading system itself with the help of FPGA technology.
10) Even some figures give the feeling that the authors presented something hastily without trying to increase the quality. For example, Figure 2 has many lines interconnected with each other. If the authors took time to improve it, they would fill the arrows with white color, and the interconnection lines (which makes it a mess) would vanish. When sending a paper for review, the authors should send their best effort that has quality, not a draft version that was written hastily. Another example is Figure 5, which has an arrow not balanced (it is very close to the right table).
Author Response- All the figures have been modified and redrawn to make them more clear as per this feedback.
11) There are no take-away messages of how exactly the HLS was performed (just stating, we used Vivado HLS is not complete). The authors should specify exactly how they used Vivado HLS, how they inserted the pragmas, how they decided on the hierarchy and the operators, and so on. In this state, no reader will benefit from it and this should not be published, because it violates the good publishing ethics of the research community (together with the very bad quality of the figures and text).
Author Response- We have added a section on HLS implementation of Order Processing block. We have described the functions, and pragmas which have been used. Top-level architecture and hierarchy have been described.
12) Even the comparisons are not well described. Just stating we used 16 cores is useless. What CPU? Why there was latency on 16 cores? Are they very old? If the authors utilized 32 cores it would be up to the speed of FPGA (so this would be better?). The comparisons should be correct and not biased. In their current form, they are biased and they do not give the full picture. Just some numbers that show that FPGA is better. But behind the numbers, I can see that it's not so simple the comparison.
Author Response – This problem has been addressed and we have provided all the details of the system on which trading system software was implemented. Also using a 32 core machine will not result in higher throughput and reduced latency since the order processing logic sequential in nature with no possibility of parallel execution (considering parallel processing overhead). Also with the increasing number of cores, the frequency of the cores in servers goes down resulting in slower execution. So, for higher throughput, higher frequency of cores is required so the 16 core machine has been used.
13) Even the algorithm used is not described correctly, and thus I cannot verify the correctness.
Author Response- Could you specify which algorithm you are referring to above?
14) Even the test measurements are not described correctly. How the testbench was created? How the validation phase passed the tests?
Author response – The test bench was written in System Verilog to generate a random mix of commands. Test bench is not self-checking testbench, it does not have any model of DUT to compare the actual response against the expected response. This is not required since the test bench measures only the latencies of execution of different commands by noting the time at which the command was delivered and time at which the response was delivered. There is no validation phase of a test case since only the latency measurements are performed. Test only checks whether the command was executed successfully or failed.
In general, the authors should prepare better this manuscript. This has many problems that prohibit the publication.
Author response- Thank you for the comments. We have taken care of this by modifying figures to make them clearer. The abstract has been modified and a section on HLS implementation has been included. We hope that this meets the requirements.
Reviewer 2 Report
This paper describes the HLS implementation of a trading system on FPGA.
The paper should be improved prior publication.
The abstract is very hard to understand and does not give enough information about the problem solved.
The description of the implemented architecture is not detailed. HLS requires annotating the C source code with suitable directives that allow the compiler to generate the hardware architecture. Authors should details how they developed the HLS version of the trading system.
Figures are stretched and/or blurred and not easy to read. Please, enlarge the figures.
References are not cited in ascending order. Moreover refs. 3 and 7 are not cited in the main text. A lot of references are websites, but the last access date is not reported. Moreover, some of the other references are older than 10 years.
Please, carefully revise the writing style of the paper.
Author Response
We would like to thank the reviewer for the comments and feedback. Please find below inline our response to the comments.
This paper describes the HLS implementation of a trading system on FPGA.
The paper should be improved prior publication.
The abstract is very hard to understand and does not give enough information about the problem solved.
Author Response – We have modified the abstract to make it clearer.
The description of the implemented architecture is not detailed. HLS requires annotating the C source code with suitable directives that allow the compiler to generate the hardware architecture. Authors should details how they developed the HLS version of the trading system.
Author Response – We have modified the manuscript and provided a section on HLS implementation. It describes architecture of HLS code, functions used, pragmas etc.
Figures are stretched and/or blurred and not easy to read. Please, enlarge the figures.
Author Response – All the figures have been modified to make them more clear
References are not cited in ascending order. Moreover refs. 3 and 7 are not cited in the main text. A lot of references are websites, but the last access date is not reported. Moreover, some of the other references are older than 10 years.
Author Response – This is taken care of. References 3 and 7 have been added. The references which are 10 year old pertain to HFT which is a major application of FPGAs in finance. We can change them later if required to include more recent references on HFT.
Please, carefully revise the writing style of the paper.
Author response – We have done modifications to the paper and added more sections.
Reviewer 3 Report
High-level synthesis is used in this article to migrate the functionality of trading system components currently implemented in software to FPGA hardware. This is done to reduce latency and increase throughput. As a result of the investigations, the authors obtained a significant improvement in the throughput of INSERT and UPDATE commands compared to the software implementation.
There are some comments and questions regarding the content of the article:
1. The authors' claim that "There have been very few instances of the use of FPGA for a complete trading system backend" (line 22, 23) looks somewhat doubtful. It would be worthwhile to consider, for example, the following works:
C. Leber, B. Geib and H. Litz, "High Frequency Trading Acceleration Using FPGAs," 2011 21st International Conference on Field Programmable Logic and Applications, 2011, pp. 317-322, doi: 10.1109/FPL.2011.64.
Y. -C. Kao, H. -A. Chen and H. -P. Ma, "An FPGA-Based High-Frequency Trading System for 10 Gigabit Ethernet with a Latency of 433 ns," 2022 International Symposium on VLSI Design, Automation and Test (VLSI-DAT), 2022, pp. 1-4, doi: 10.1109/VLSI-DAT54769.2022.9768065.
In addition, the literature used in the work is not sufficiently new, in particular [5], [6], [7], [26].
2. Lines 61-62: What software or framework did you compare your solution to?
3. Figure 4: What interface is used for the order DB connection? Is it just Register File?
4. The case where the next commands (INSERT, UPDATE. INSERT, UPDATE ...) is not considered. What is the latency in this case? How will run pipeline execution of the UPDATE command?
5. There are no references to literary sources [3], [7]. [23].
6. Literary sources are not placed in the order of appearance of references in the text. This makes the article difficult to read.
7. There are grammatical errors in the text of the article, in particular, in line 136, the sentence begins with a lowercase letter; the year of literary sources 1 and 3 is not indicated (2007 and 2008, respectively).
Author Response
We would like to thank the reviewer for the suggestions and feedback. Please find our response inline to the comments below:
High-level synthesis is used in this article to migrate the functionality of trading system components currently implemented in software to FPGA hardware. This is done to reduce latency and increase throughput. As a result of the investigations, the authors obtained a significant improvement in the throughput of INSERT and UPDATE commands compared to the software implementation.
There are some comments and questions regarding the content of the article:
- The authors' claim that "There have been very few instances of the use of FPGA for a complete trading system backend" (line 22, 23) looks somewhat doubtful. It would be worthwhile to consider, for example, the following works:
- Leber, B. Geib and H. Litz, "High Frequency Trading Acceleration Using FPGAs," 2011 21st International Conference on Field Programmable Logic and Applications, 2011, pp. 317-322, doi: 10.1109/FPL.2011.64.
Author Response - The article mentioned above refers to the acceleration of High-Frequency Trading (HFT). The HFT system is located outside a stock exchange and is used by the users submitting trade requests to a trading system. In our study, we are trying to accelerate the trading system component itself (order processing block) which is located inside the stock exchange. Both systems are different and cannot be compared. BTW, an HFT system is used by stock exchange users to generate a large number of orders in a fraction of a second, based on market inputs. It is used by the users submitting Buy and Sell orders to the stock exchange.
- -C. Kao, H. -A. Chen and H. -P. Ma, "An FPGA-Based High-Frequency Trading System for 10 Gigabit Ethernet with a Latency of 433 ns," 2022 International Symposium on VLSI Design, Automation and Test (VLSI-DAT), 2022, pp. 1-4, doi: 10.1109/VLSI-DAT54769.2022.9768065.
Author Response- As mentioned above, this article also refers to HFT system which is located outside a stock exchange. We are trying to accelerate the trading system itself which sits inside the premises of a stock exchange.
In addition, the literature used in the work is not sufficiently new, in particular [5], [6], [7], [26].
Author Response- After an extensive literature survey, we have not found any designs implementing trading systems backend with FPGA. References 5, 6, 7 are related to HLS flow which we have used. We studied them to understand the HLS flow which we have used for implementation. Reference 26 is for HFT mentioned above. We can site more recent references if required. However, we have not found any literature on the acceleration of the trading system itself using an FPGA.
- Lines 61-62: What software or framework did you compare your solution to?
Author Response -As mentioned above, we could not find any literature on acceleration of trading system itself using FPGA. So we compared our solution to software implementation of trading system of a major stock exchange in India. This exchange is our client. The latency and throughput numbers are taken from the stock exchange.
- Figure 4: What interface is used for the order DB connection? Is it just Register File?
Author Response- The order data base is implemented with Sram and SRAM controller has an AXI-4 interface.
- The case where the next commands (INSERT, UPDATE. INSERT, UPDATE ...) is not considered. What is the latency in this case? How will run pipeline execution of the UPDATE command?
Author Response- We have generated a random mix of different command patterns. So this sequence would have occurred with high probability. Anyways latency of INSERT or UPDATE is not affected by the next command. Only when a particular command has been completely executed and data committed to the memory, we take up the next command for execution. There cannot be a pipelined execution of INSERT followed by UPDATE i.e. We cannot start UPDATE execution when the previous INSERT is still in the process of getting executed.
- There are no references to literary sources [3], [7]. [23].
Author Response- References to resources 3 and 7 are included. [23] has been removed.
- Literary sources are not placed in the order of the appearance of references in the text. This makes the article difficult to read.
Author response – We understand this is a problem. However, other reviewers have asked questions specific to references. So, if the reference numbers are changed in a major way now, they will find it difficult to review the response related to comments. We will take care of this in the final version of the paper if selected for publication.
- There are grammatical errors in the text of the article, in particular, in line 136, the sentence begins with a lowercase letter; the year of literary sources 1 and 3 is not indicated (2007 and 2008, respectively).
Author Response – These have been corrected in the revised manuscript
Reviewer 4 Report
In the paper the authors describe the design of a financial trading system backend using FPGA implemented with High-Level Synthesis (HLS) flow.
The FPGA implementation reduces the latency of order processing and increases the throughput of the trading system as compared to software implementation.
They use HLS to migrate to FPGA hardware the functionality of trading system components which are currently implemented in software.
Text is well written and well organized.
Section 3 clearly describes the system architecture.
Line 236 is broken. Its paragraph is not justified.
Define DRAM and BRAM.
Section 5, what happens when you ran out of memory (order database,
order nodes) because of too many orders ?
Define DUT.
Regarding the design of the FPGA solution, any other design solution was
considered ? I suppose there are many ways to deal with the stream of orders, with respect to architectural components and connections. Any alternative to the design described in Section 3 was ever considered ?
Section 7 presents the pipelined execution of the UPDATE command. A pipeline of two stages is used. Is there any condition when the Fetch stage and the Execute stage conflict, i.e., there is a dependence between the Execute stage of order "n" and the Fetch stage of order "n+1", so as to stall the pipeline ?
"With HLS, we achieve almost 4X-5X improvement in throughput for INSERT and UPDATE commands compared to software implementation."
But nothing is said about the software implementation. Which data structures and synchronization mechanisms (multicore implementation) were used ?
Author Response
We would like to thank the reviewer for the suggestions and feedback. Please find our response inline to the comments below:
In the paper the authors describe the design of a financial trading system backend using FPGA implemented with High-Level Synthesis (HLS) flow.
The FPGA implementation reduces the latency of order processing and increases the throughput of the trading system as compared to software implementation.
They use HLS to migrate to FPGA hardware the functionality of trading system components which are currently implemented in software.
Text is well written and well organized.
Section 3 clearly describes the system architecture.
Line 236 is broken. Its paragraph is not justified.
Author Response- This has been corrected.
Define DRAM and BRAM.
Author Response- This has been defined.
Section 5, what happens when you ran out of memory (order database,
order nodes) because of too many orders ?
Author Response- At present, the order database which is implemented in Static Ram is used only up to 10% capacity. (stock exchange is receiving upto 100,000 orders while it is provisioned for 1 million orders). Even if we run out of capacity (receive more than 1 million orders), we can store the orders in on-chip high bandwidth memory (HBM). We have 16Gbytes of HBM available on FPGA. The only problem with this will be – HBM is not high-speed memory (it is similar to Dynamic Ram) so, the order processing rate will be less and latency will increase.
Define DUT.
Author Response- DUT stands for “design under test”. This has been added in the manuscript.
Regarding the design of the FPGA solution, any other design solution was
considered ? I suppose there are many ways to deal with the stream of orders, with respect to architectural components and connections. Any alternative to the design described in Section 3 was ever considered ?
Author Response- We are currently working on a design which uses a low latency DRAM (LLDRAM) instead of static Ram for storing order data base.
Section 7 presents the pipelined execution of the UPDATE command. A pipeline of two stages is used. Is there any condition when the Fetch stage and the Execute stage conflict, i.e., there is a dependence between the Execute stage of order "n" and the Fetch stage of order "n+1", so as to stall the pipeline ?
Author Response- There is no such dependence and possibility of pipeline stall does not exist.
"With HLS, we achieve almost 4X-5X improvement in throughput for INSERT and UPDATE commands compared to software implementation."
But nothing is said about the software implementation. Which data structures and synchronization mechanisms (multicore implementation) were used ?
Author Response- Data structures used in software are different since there is no consideration of Block RAM for software implementation. The software uses hashmap and treemap data structures for order book. As for the synchronization, single-writer principles have been followed to avoid locking contentions in the performance-critical path. In a few scenarios, Compare and Swap low latency locks are used. Dedicated isolated cores are assigned to every process to avoid CPU switching.
Round 2
Reviewer 1 Report
In this revised version the manuscript is improved. It is more professional and presents many desired details. I have some minor remarks to report:
1) The authors mention HLS and Verilog implementation. I would like to see a comparison table between them also (not only with software) and also a discussion on why the HLS is not as good as the Verilog implementation (for example did it utilize a finite state machine with many more states?)
2) I would like to see a pseudo-algorithm of the processing flow for implemented processing block. The authors describe the operations (like INSERT or DELETE) but a picture with a flow will contribute more to the understanding.
3) A motivational sentence should be inserted in the abstract, as an attention gatherer.
4) In Figure 1, I propose the Order processing block to have a different background (not gray) to denote that the authors are working on accelerating this part. Also, in the caption, the authors should say "With green color, we denote the part that we have accelerated" or something similar. Conveying information using multiple channels is desired in a scientific paper.
5) In Figure 5, there is "=k", which I think is wrong, or the figure does not convey the correct information. Utilizing a larger caption here will help the reader understand what he sees.
6) @361. The authors mention Intel 3Ghz, but this is not the formal way to report the CPU. They should give the CPU model exactly, for example, Intel Atom® Processor Z3795.
7) The authors do not give details on the software implementation. It is a commercial solution? What is the name? How it was implemented? Does it use threads or processes? Is it scalable? It is an old implementation or a new one? The authors compare their solution with this software, so we have to know as much as possible about this software.
8) I would like to see a table with all HLS parameters used in the design. The authors mention some pragmas in the text, but a table reporting all HLS pragmas and parameters would give a better picture of the HLS implementation. Also, the HLS implementation took 1.2 months for experienced designers with HLS or for engineers that just started HLS? The level of knowledge is important when stating such information.
The paper is improved and the authors are working on the correct way to provide a sound and interesting scientific paper.
Author Response
Reviewer 1
We would like to thank the reviewer for his comments and feedback. Please find our response to the comments inline below:
In this revised version the manuscript is improved. It is more professional and presents many desired details. I have some minor remarks to report:
1) I would like to see a pseudo-algorithm of the processing flow for implemented processing block. The authors describe the operations (like INSERT or DELETE) but a picture with a flow will contribute more to the understanding.
Author Response – This has been added. Please refer to figure 8
2) A motivational sentence should be inserted in the abstract, as an attention gatherer.
Author Response - This has been done. We have added a sentence “our design is one of the first attempts to speed up the order processing in a trading system using FPGA technology and HLS flow”
3) In Figure 1, I propose the Order processing block to have a different background (not gray) to denote that the authors are working on accelerating this part. Also, in the caption, the authors should say "With green color, we denote the part that we have accelerated" or something similar. Conveying information using multiple channels is desired in a scientific paper.
Author Response – This has been done. The order processing block is shown in green colour.
4) In Figure 5, there is "=k", which I think is wrong, or the figure does not convey the correct information. Utilizing a larger caption here will help the reader understand what he sees.
Author Response – This has been corrected. We have removed “=” sign. Now figure means -if the Orders 0, 1, 2..k have order IDs m0, m1, …mk, then at the address offset mk, integer k is stored. (the Order0 is stored at offset 0, order1 is stored at offset 1, order k is stored at offset k. The orders have OrderIDs of m0, m1, m2, mk respectively. So mk is the address to the OrderIndex table and at address mk, k is stored (which is the offset of the order k, in the order book data base)
5) @361. The authors mention Intel 3Ghz, but this is not the formal way to report the CPU. They should give the CPU model exactly, for example, Intel Atom® Processor Z3795.
Author Response- It is Intel (R) Xenon(R) CPU ES-2667 V3 @ 3.20 GHz
6) The authors do not give details on the software implementation. It is a commercial solution? What is the name? How it was implemented? Does it use threads or processes? Is it scalable? It is an old implementation or a new one? The authors compare their solution with this software, so we have to know as much as possible about this software.
Author Response- Software has been developed by our company (Tata Consultancy Services) for a major stock exchange. It is a commercial solution and it will not be possible to disclose the name since the software is for a particular client and client's name cannot be disclosed. The software is however multi-threaded and below are the details :
- Data structures used in the software are different since there is no consideration of Block RAM for software implementation. The software uses hashmap and treemap data structures for the order book. As for the synchronization, single-writer principles have been followed to avoid locking contentions in the performance-critical path. In a few scenarios, Compare and Swap low latency lock are used. Dedicated isolated cores are assigned to every process to avoid CPU switching.
We have added these details in the manuscript also.
7) I would like to see a table with all HLS parameters used in the design. The authors mention some pragmas in the text, but a table reporting all HLS pragmas and parameters would give a better picture of the HLS implementation. Also, the HLS implementation took 1.2 months for experienced designers with HLS or for engineers that just started HLS? The level of knowledge is important when stating such information.
Author Response – This has been added. We have added experience level of HLS engineers in manuscript
8) The authors mention HLS and Verilog implementation. I would like to see a comparison table between them also (not only with software) and also a discussion on why the HLS is not as good as the Verilog implementation (for example did it utilize a finite state machine with many more states?)
Author Response – We have not given a comparison with Verilog implementation as the results were similar. In fact, all the test cases run on the HLS version of the design were run on software and there was only 10-20% difference in the performance numbers for various commands. The HLS implementation was almost as efficient as Verilog. The reason for this may be the pragmas used in HLS and pipelined design. The major difference with Verilog implementation was the time taken. It took almost 6 months for 4 experienced engineers to write Verilog code. Since the HLS implementation gave us very satisfactory results, it was delivered to our client and no further work was done on the Verilog implementation.
The paper is improved and the authors are working on the correct way to provide a sound and interesting scientific paper.
Reviewer 2 Report
Authors addressed all my comments.
Author Response
Reviewer 2
Comments and Suggestions for Authors
The authors addressed all my comments.
Author response - We would like to thank the reviewer for the feedback and comments.